# Learning search spaces for Bayesian optimization: Another view of hyperparameter transfer learning

**Valerio Perrone, Huibin Shen, Matthias Seeger, Cédric Archambeau, Rodolphe Jenatton**[*]
Amazon
Berlin, Germany
{vperrone, huibishe, matthis, cedrica}@amazon.com

## Abstract

Bayesian optimization (BO) is a successful methodology to optimize black-box functions that are expensive to evaluate. While traditional methods optimize each black-box function in isolation, there has been recent interest in speeding up BO by transferring knowledge across multiple related black-box functions. In this work, we introduce a method to automatically design the BO *search space* by relying on evaluations of previous black-box functions. We depart from the common practice of defining a set of arbitrary search ranges a priori by considering search space geometries that are learned from historical data. This simple, yet effective strategy can be used to endow many existing BO methods with transfer learning properties. Despite its simplicity, we show that our approach considerably boosts BO by reducing the size of the search space, thus accelerating the optimization of a variety of black-box optimization problems. In particular, the proposed approach combined with random search results in a parameter-free, easy-to-implement, robust hyperparameter optimization strategy. We hope it will constitute a natural baseline for further research attempting to warm-start BO.

## 1 Introduction

Tuning the hyperparameters (HPs) of machine leaning (ML) models and in particular deep neural networks is critical to achieve good predictive performance. Unfortunately, the mapping of the HPs to the prediction error is in general a black-box in the sense that neither its analytical form nor its gradients are available. Moreover, every (noisy) evaluation of this black-box is time-consuming as it requires retraining the model from scratch. Bayesian optimization (BO) provides a principled approach to this problem: an acquisition function, which takes as input a cheap probabilistic surrogate model of the target black-box function, repeatedly scores promising HP configurations by performing an explore-exploit trade-off [30, 22, 37]. The surrogate model is built from the set of black-box function evaluations observed so far. For example, a popular approach is to impose a Gaussian process (GP) prior on the unobserved target black-box function $f(\mathbf{x})$. Based on a set of evaluations $\{f(\mathbf{x}^i)\}_{i=1}^n$, possibly perturbed by Gaussian noise, one can compute the posterior GP, characterized by a posterior mean and a posterior (co)variance function. The next query points are selected by optimizing an acquisition function, such as the expected improvement [30], which is analytically tractable given these two quantities. While BO takes the human out of the loop in ML by automating HP optimization (HPO), it still requires the user to define a suitable search space a priori. Defining a default search space for a particular ML problem is difficult and left to human experts.

In this work, we automatically design the BO search space, which is a critical input to any BO procedure applied to HPO, based on historical data. The proposed approach relies on the observation that HPO problems occurring in ML are often related (for example, tuning the HPs of an ML model

---

[*]Work done while affiliated with Amazon; now at Google Brain, Berlin, rjenatton@google.com

trained on different data sets [43, 3, 51, 35, 11, 32, 9, 28]). Moreover, our method learns a suitable *search space* in a universal fashion: it can endow *any* BO algorithm with transfer learning capabilities. For instance, we demonstrate this feature with three widely used HPO algorithms – random search [4], SMAC [20] and hyperband [29]. Further, we investigate the use of novel geometrical representations of the search spaces, departing from the traditional rectangular boxes. In particular, we show that an ellipsoidal representation is not only simple to compute and manipulate, but leads to faster black-box optimization, especially as the dimension of the search space increases.

## 2 Related work and contributions

Previous work has implemented BO transfer learning in many different ways. For instance, the problem can be framed as a multi-task learning problem, where each run of BO corresponds to a task. Tasks can be modelled jointly or as being conditionally independent with a multi-output GP [43], a Bayesian neural network [41], a multi-layer perceptron with Bayesian linear regression heads [39, 32], possibly together with some embedding [28], or a weighted combination of GPs [35, 9]. Alternatively, several authors attempted to rely on manually defined meta-features in order to measure the similarity between BO problems [3, 51, 35]. If these problems further come in a specific ordering (e.g., because of successive releases of an ML model), the successive surrogate models can be fit to the residuals relative to predictions of the previous learned surrogate model [15, 33]. In particular, if GP surrogates are used, the new GP is centered around the predictive mean of the previously learned GP surrogate. Finally, rather than fitting a surrogate model to all past data, some transfer can be achieved by warm-starting BO with the solutions to the previous BO problems [10, 50].

The work most closely related to ours is [49], where the search space is pruned during BO, removing unpromising regions based on information from related BO problems. Similarity scores between BO problems are computed from data set meta-features. While we also aim to restrict the BO search space, our approach is different in many ways. First, we do not require meta-features, which in practice can be hard to obtain and need careful manual design. Second, our procedure works completely offline, as a preprocessing step, and does not require feedback from the black-box function being optimized. Third, it is parameter-free and model-free. By contrast, [49] rely on a GP model and have to select a radius and the fraction of the space to prune. Finally, [49] use a discretization step to prune the search space, which may not scale well as its dimension increases. The generality of our approach is such that [49] could be used on top of our proposed method (while the converse is not true).

Another line of research has developed search space expansion strategies for BO. Those approaches are less dependent on the initial search space provided by the users, incrementally *expanding* it during BO [36, 31]. None of this research has considered transfer learning. A related idea to learn hyperparameter importance has been explored in [45], where a post-hoc functional ANOVA analysis is used to learn priors over the hyperparameter space. Again, such techniques could be used together with our approach, which would define the initial search space in a data driven manner.

Our contributions are: (1) We introduce a simple and generic class of methods that design compact search spaces from historical data, making it possible to endow any BO method with transfer learning properties, (2) we explore and demonstrate the value of new geometrical representations of search spaces beyond the rectangular boxes traditionally employed, and (3) we show over a broad set of transfer learning experiments that our approach consistently boosts the performance of the optimizers it is paired with. When combined with random search, the resulting simple and parameter-free optimization strategy constitutes a strong baseline, which we hope will be adopted in future research.

## 3 Black-box function optimization on a reduced search space

Consider $T + 1$ black-box functions $\{f_t(\cdot)\}_{t=0}^T$, defined on a common search space $\mathcal{X} \subseteq \mathbb{R}^p$. The functions are expensive-to-evaluate, possibly non-convex, and accessible only through their values, without gradient information. In loose terms, the functions $\{f_t(\cdot)\}_{t=0}^T$ are assumed *related*, corresponding for instance to the evaluations of a given ML model over $T + 1$ data sets (in which case $\mathcal{X}$ is the set of feasible HPs of this model). Our goal is to minimize $f_0$:

$$\min_{\mathbf{x} \in \mathcal{X}} f_0(\mathbf{x}). \tag{1}$$

However, for $t \geq 1$, we have access to $n_t$ noisy evaluations of the function $f_t$, which we denote by $\mathcal{D}_t = \{(\mathbf{x}_{i,t}, y_{i,t})\}_{i=1}^{n_t}$. In this work, we consider methods that take the previous evaluations $\{\mathcal{D}_t\}_{t=1}^T$

as inputs, and output a search space $\hat{\mathcal{X}} \subseteq \mathcal{X}$, so that we solve the following problem instead of (1):

$$\min_{\mathbf{x} \in \hat{\mathcal{X}}} f_0(\mathbf{x}). \tag{2}$$

The local minima of (2) are a subset of the local minima of (1). Since $\hat{\mathcal{X}}$ is more compact (formally defined later), BO methods will find those minima faster, i.e., with fewer function evaluations. Hence, we aim to design $\hat{\mathcal{X}}$ such that it contains a "good" set of local minima, close to the global ones of $\mathcal{X}$.

## 4 Data-driven search space design via transfer learning

**Notations and preliminaries.** We define $(\mathbf{x}_t^\star, y_t^\star)$ as the element in $\mathcal{D}_t$ that reaches the smallest (i.e., best) evaluation for the black-box $t$, i.e., $(\mathbf{x}_t^\star, y_t^\star) = \arg\min_{(\mathbf{x}_t, y_t) \in \mathcal{D}_t} y_t$. For any two vectors $\mathbf{u}$ and $\mathbf{w}$ in $\mathbb{R}^p$, $\mathbf{u} \le \mathbf{w}$ stands for the element-wise inequalities $u_j \le w_j$ for $j \in \{1, \ldots, p\}$. We also denote by $|\mathbf{u}|$ the vector with entries $|u_j|$ for $j \in \{1, \ldots, p\}$. For a collection $\{\mathbf{w}_t\}_{t=1}^T$ of $T$ vectors in $\mathbb{R}^p$, we denote by $\min\{\mathbf{w}_t\}_{t=1}^T$, respectively $\max\{\mathbf{w}_t\}_{t=1}^T$, the $p$-dimensional vector resulting from the element-wise minimum, respectively maximum, over the $T$ vectors. Finally, for any symmetric matrix $\mathbf{A} \in \mathbb{R}^{p \times p}$, $\mathbf{A} \succ \mathbf{0}$ indicates that $\mathbf{A}$ is positive definite.

We assume that the original search space $\mathcal{X}$ is defined by axis-aligned ranges that can be thought of as a bounding box: $\mathcal{X} = \{\mathbf{x} \in \mathbb{R}^p | \, \mathbf{l}_0 \le \mathbf{x} \le \mathbf{u}_0\}$ where $\mathbf{l}_0$ and $\mathbf{u}_0$ are the initial vectors of lower and upper bounds. Search spaces represented as boxes are commonly used in popular BO packages such as Spearmint [38], GPyOpt [1], GPflowOpt [27] , Dragonfly [24] and Ax [7].

The methodology we develop applies to numerical parameters (either integer or continuous). If the problems under consideration exhibit categorical parameters (we have such examples in our experiments, Section 6), we let $\mathcal{X} = \mathcal{X}_{\text{cat}} \times \mathcal{X}_{\text{num}}$. Our methodology then applies to $\mathcal{X}_{\text{num}}$ only, keeping $\mathcal{X}_{\text{cat}}$ unchanged. Hence, in the remainder the dimension $p$ refers to the dimension of $\mathcal{X}_{\text{num}}$.

### 4.1 Search space estimation as an optimization problem

The reduced search space $\hat{\mathcal{X}}$ we would like to learn is defined by a parameter vector $\boldsymbol{\theta} \in \mathbb{R}^k$. To estimate $\hat{\mathcal{X}}$, we consider the following constrained optimization problem:

$$\min_{\boldsymbol{\theta} \in \mathbb{R}^k} \mathcal{Q}(\boldsymbol{\theta}) \text{ such that for } t \ge 1, \, \mathbf{x}_t^\star \in \hat{\mathcal{X}}(\boldsymbol{\theta}), \tag{3}$$

where $\mathcal{Q}(\boldsymbol{\theta})$ is some measure of volume of $\hat{\mathcal{X}}(\boldsymbol{\theta})$; concrete examples are given in Sections 4.2 and 4.3. In solving (3), we find a search space $\hat{\mathcal{X}}(\boldsymbol{\theta})$ that contains all solutions $\{\mathbf{x}_t^\star\}_{t=1}^T$ to previously solved black-box optimization problems, while at the same time minimizing $\mathcal{Q}(\boldsymbol{\theta})$. Note that $\hat{\mathcal{X}}(\boldsymbol{\theta})$ can only get larger (as measured by $\mathcal{Q}(\boldsymbol{\theta})$) as more related black-box optimization problems are considered. Moreover, formulation (3) does not explicitly use the $y_t^\star$'s and never compares them across tasks. As a result, unlike previous work such as [51, 9], we need not normalize the tasks (e.g., whitening).

### 4.2 Search space as a low-volume bounding box

The first instantiation of (3) is a search space defined by a bounding box (or hyperrectangle), which is parameterized by the lower and upper bounds $\mathbf{l}$ and $\mathbf{u}$. More formally, $\hat{\mathcal{X}}(\boldsymbol{\theta}) = \{\mathbf{x} \in \mathbb{R}^p | \, \mathbf{l} \le \mathbf{x} \le \mathbf{u}\}$ and $\boldsymbol{\theta} = (\mathbf{l}, \mathbf{u})$, with $k = 2p$. A tight bounding box containing all $\{\mathbf{x}_t^\star\}_{t=1}^T$ can be obtained as the solution to the following constrained minimization problem:

$$\min_{\mathbf{l} \in \mathbb{R}^p, \, \mathbf{u} \in \mathbb{R}^p} \frac{1}{2} \|\mathbf{u} - \mathbf{l}\|_2^2 \text{ such that for } t \ge 1, \, \mathbf{l} \le \mathbf{x}_t^\star \le \mathbf{u} \tag{4}$$

where the compactness of the search space is enforced by a squared $\ell_2$ term that penalizes large ranges in each dimension. This problem has a simple closed-form solution $\boldsymbol{\theta}_b^* = (\mathbf{l}^*, \mathbf{u}^*)$, where

$$\mathbf{l}^* = \min\{\mathbf{x}_t^\star\}_{t=1}^T \quad \text{and} \quad \mathbf{u}^* = \max\{\mathbf{x}_t^\star\}_{t=1}^T. \tag{5}$$

These solutions are simple and intuitive: while the initial lower and upper bounds $\mathbf{l}_0$ and $\mathbf{u}_0$ may define overly wide ranges, the new ranges of $\hat{\mathcal{X}}(\boldsymbol{\theta}_b^*)$ are the smallest ranges containing all the

related solutions $\{\mathbf{x}_t^\star\}_{t=1}^T$. The resulting search space $\hat{\mathcal{X}}(\boldsymbol{\theta}_b^*)$ defines a new, tight bounding box that can directly be used with any optimizer operating on the original $\mathcal{X}$. Despite the simplicity of the definition of $\hat{\mathcal{X}}(\boldsymbol{\theta}_b^*)$, we show in Section 6 that this approach constitutes a surprisingly strong baseline, even when combined with random search only. We will generalize the optimization problem (4) in Section 5 to obtain solutions that account for outliers contained $\{\mathbf{x}_t^\star\}_{t=1}^T$ and, as a result, produce an even tighter search space.

### 4.3 Search space as a low-volume ellipsoid

The second instantiation of (3) is a search space defined by a hyperellipsoid (i.e., affine transformations of a unit $\ell_2$ ball), which is parameterized by a symmetric positive definite matrix $\mathbf{A} \in \mathbb{R}^{p \times p}$ and an offset vector $\mathbf{b} \in \mathbb{R}^p$. More formally, $\hat{\mathcal{X}}(\boldsymbol{\theta}) = \{\mathbf{x} \in \mathbb{R}^p \mid \|\mathbf{A}\mathbf{x} + \mathbf{b}\|_2 \leq 1\}$ and $\boldsymbol{\theta} = (\mathbf{A}, \mathbf{b})$, with $k = p(p+3)/2$. Using the classical Löwner-John formulation [21], the lowest volume ellipsoid covering all points $\{\mathbf{x}_t^\star\}_{t=1}^T$ is the solution to the following problem (see Section 8.4 in [5]):

$$\min_{\mathbf{A} \in \mathbb{R}^{p \times p}, \, \mathbf{A} \succ \mathbf{0}, \, \mathbf{b} \in \mathbb{R}^p} \log \det(\mathbf{A}^{-1}) \text{ such that for } t \geq 1, \, \|\mathbf{A}\mathbf{x}_t^\star + \mathbf{b}\|_2 \leq 1, \qquad (6)$$

where the $T$ norm constraints enforce $\mathbf{x}_t^\star \in \hat{\mathcal{X}}(\boldsymbol{\theta})$, while the minimized objective is a strictly increasing function of the volume of the ellipsoid $\propto 1/\sqrt{\det(\mathbf{A})}$ [17]. This problem is convex, admits a unique solution $\boldsymbol{\theta}_e^* = (\mathbf{A}^*, \mathbf{b}^*)$, and can be solved efficiently by interior-points algorithms [42]. In our experiments, we use CVXPY [6].

Intuitively, an ellipsoid should be more suitable than a hyperrectangle when the points $\{\mathbf{x}_t^\star\}_{t=1}^T$ we want to cover do not cluster in the corners of the box. In Section 6, a variety of real-world ML problems suggest that the distribution of the solutions $\{\mathbf{x}_t^\star\}_{t=1}^T$ supports this hypothesis. We will also generalize the optimization problem (6) in Section 5 to obtain solutions that account for outliers contained in $\{\mathbf{x}_t^\star\}_{t=1}^T$.

### 4.4 Optimizing over ellipsoidal search spaces

We cannot directly plug ellipsoidal search spaces into standard HPO procedures. Algorithm 1 details how to adapt random search, and as a consequence also methods like hyperband [29], to an ellipsoidal search space. In a nutshell, we use rejection sampling to guarantee uniform sampling in $\mathcal{X} \cap \hat{\mathcal{X}}(\boldsymbol{\theta}_e^*)$: we first sample uniformly in the $p$-dimensional ball, then apply the inverse mapping of the ellipsoid [12], and finally check whether the sample belongs to $\mathcal{X}$. The last step is important as not all points in the ellipsoid may be valid points in $\mathcal{X}$. For example, a HP might be restricted to only take positive values. However, after fitting the ellipsoid, some small amount of its volume might include negative values. Finally, ellipsoidal search spaces cannot directly be used with more complex, model-based BO engines, such as GPs. They would require resorting to constrained BO modelling [16, 13, 14], e.g., to optimize the acquisition function over the ellipsoid, which would add significant complexity to the procedure. Hence, we defer this investigation to future work.

---

**Algorithm 1** Rejection sampling algorithm to uniformly sample in an ellipsoidal search space

---

1: **procedure** ELLIPSOIDRANDOMSAMPLING($\boldsymbol{\theta}_e^*, \mathcal{X}$)  $\qquad \triangleright \boldsymbol{\theta}_e^*$ is the solution from Section 4.3
2: $\qquad \mathbf{A}^*, \mathbf{b}^* \leftarrow \boldsymbol{\theta}_e^*$ and IS_FEASIBLE $\leftarrow$ FALSE
3: $\qquad$ **while** not IS_FEASIBLE **do**
4: $\qquad\qquad \mathbf{z} \sim \mathcal{N}(\mathbf{0}, \mathbf{I})$, with $\mathbf{z} \in \mathbb{R}^p$, and $r \sim \mathcal{U}(0, 1)$
5: $\qquad\qquad \mathbf{t} \leftarrow \frac{r^{1/p}}{\|\mathbf{z}\|_2} \mathbf{z}$  $\qquad\qquad\qquad\qquad \triangleright \mathbf{t}$ is uniformly distributed in the unit $\ell_2$ ball [19]
6: $\qquad\qquad \mathbf{x} \leftarrow (\mathbf{A}^*)^{-1}(\mathbf{t} - \mathbf{b}^*)$  $\qquad\quad \triangleright \mathbf{x}$ is uniformly distributed in the ellipsoid $\hat{\mathcal{X}}(\boldsymbol{\theta}_e^*)$ [12]
7: $\qquad\qquad$ **if** $\mathbf{x} \in \mathcal{X}$ **then**  $\qquad\qquad \triangleright$ We check if $\mathbf{x}$ is valid since we may not have $\hat{\mathcal{X}}(\boldsymbol{\theta}_e^*) \subseteq \mathcal{X}$
8: $\qquad\qquad\qquad$ IS_FEASIBLE $\leftarrow$ TRUE
9: $\qquad$ **return** $\mathbf{x}$

---

## 5 Handling outliers in the historical data

The search space chosen by our method is the smallest hyperrectangle or hyperellipsoid enclosing a set of solutions $\{\mathbf{x}_t^\star\}_{t=1}^T$ found by optimizing related black-box optimization problems. In order

to exploit as much information as possible, a large number of related problems may be considered. However, the learned search space volume might increase as a result, which will make black-box optimization algorithms, such as BO, less effective. For example, if some of these problems depart significantly from the other black-box optimization problems, their contribution to the volume increase might be disproportionate and discarding them will be beneficial. In this section, we extend of our methodology to exclude such outliers automatically.

We allow for some $\mathbf{x}_t^\star$ to violate feasibility, but penalize such violations by way of slack variables. To exclude outliers from the hyperrectangle, problem (4) is modified as follows:

$$\min_{\substack{\mathbf{l}\in\mathbb{R}^p,\ \mathbf{u}\in\mathbb{R}^p,\ \xi_t^-\geq 0,\ \xi_t^+\geq 0 \\ \text{for } t\geq 1,\ \mathbf{l}-\xi_t^-\,|\mathbf{l}_0|\leq \mathbf{x}_t^\star\leq \mathbf{u}+\xi_t^+\,|\mathbf{u}_0|}} \frac{\lambda_b}{2}\|\mathbf{u}-\mathbf{l}\|_2^2 + \frac{1}{2T}\sum_{t=1}^T(\xi_t^- + \xi_t^+), \tag{7}$$

where $\lambda_b \geq 0$ is a regularization parameter, and $\{\xi_t^-\}_{t=1}^T$ and $\{\xi_t^+\}_{t=1}^T$ the slack variables associated respectively to $\mathbf{l}$ and $\mathbf{u}$, which we make scale-free by using $|\mathbf{l}_0|$ and $|\mathbf{u}_0|$. Slack variables can also be used to exclude outliers from an ellipsoidal search region [26, 42] by rewriting (6) as follows:

$$\min_{\substack{\mathbf{A}\in\mathbb{R}^{p\times p},\ \mathbf{A}\succ\mathbf{0},\ \mathbf{b}\in\mathbb{R}^p,\ \xi_t\geq 0 \\ \text{for } t\geq 1,\ \|\mathbf{A}\mathbf{x}_t^\star+\mathbf{b}\|_2\leq 1+\xi_t}} \lambda_e \log\det(\mathbf{A}^{-1}) + \frac{1}{T}\sum_{t=1}^T \xi_t. \tag{8}$$

where $\lambda_e \geq 0$ is a regularization parameter and $\{\xi_t\}_{t=1}^T$ the slack variables . Note that the original formulations (4) and (6) are recovered when $\lambda_b$ or $\lambda_e$ tend to zero, as the optimal solution is then found when all the slack variables are equal to zero. By contrast, when $\lambda_b$ or $\lambda_e$ get larger, more solutions in the set $\{\mathbf{x}_t^\star\}_{t=1}^T$ are ignored, leading to a tighter search space.

To set $\lambda_b$ and $\lambda_e$, we proceed in two steps. First, we compute the optimal solution $\mathcal{Q}(\theta^*)$ of the original problem, namely (4) and (6) for the bounding box and ellipsoid, respectively. $\mathcal{Q}(\theta^*)$ captures the scale of the problem at hand. Then, we look at $\lambda = s/\mathcal{Q}(\theta^*)$ for $s$ in a small, scale-free, grid of values and select the smallest value of $\lambda$ that leads to no more than $(1-\nu)\times T$ solutions from $\{\mathbf{x}_t^\star\}_{t=1}^T$ (by checking the number of active, i.e., strictly positive, slack variables in (7) and (8)). We therefore turn the selection of the abstract regularization parameter $\lambda$ to the more interpretable choice of $\nu$ as a fraction of outliers. In our experiments, we purely determined those values on the toy SGD synthetic setting (Section 6.1), and then applied it as a default to the real-world problems with no extra tuning (this led to $\nu_b = 0.5$ and $\nu_e = 0.1$).

## 6 Experiments

Our experiments are guided by three key messages. First, our method can be combined with a large number of HPO algorithms. Hence, we obtain a modular design for HPO (and BO) which is convenient when building ML systems. Second, by introducing parametric assumptions (with the box and the ellipsoid), we show empirically that our approach is more robust to low-data regimes compared to model-based approaches. Third, our simple method induces transfer learning by reducing the search space of BO. The method compares favorably to more complex alternative models for transfer learning proposed in the literature, thus setting a competitive baseline for future research.

In the experiments, we consider combinations of search space definitions and HPO algorithms. `Box` and `Ellipsoid` refer to learned hyperrectangular (Section 4.2) and hyperellipsoidal search spaces (Section 4.3). When none of these prefixes are used, we defined the search space using manually specified bounding boxes (see Subsections 6.1–6.3 for the specifics). We further consider a diverse set of HPO algorithms: random search, Hyperband [29], GP-based BO, GP-based BO with input warping [40], random forest-based BO [20], and adaptive bayesian linear regression-based BO [32], which are denoted respectively by `Random`, `HB`, `GP`, `GP warping`, `SMAC`, and `ABLR`. We assessed the transfer learning capabilities in a leave-one-task-out fashion, meaning that we leave out one of the black-box optimization problems and then aggregate the results. Each curve and shaded area in the plots respectively corresponds to the mean metric and standard error obtained over 10 independent replications times the number of leave-one-task-out runs. To report the results and ease comparisons of tasks, we normalize the performance curves of each model by the best value obtained by random search, inspired by [15, 44]. Consequently, each random search performance curve ends up at 1 (or 0 when the metric is log transformed).

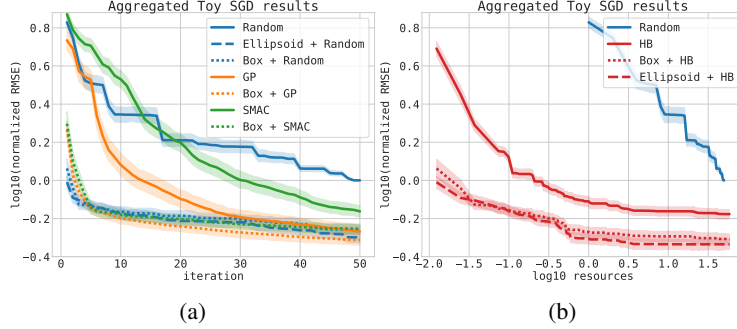

(a)                 (b)

Figure 1: Tuning SGD for ridge regression. (a) Comparison of BO algorithms with `Box` transfer learning counterparts. (b) Comparison of resource-aware BO with transfer learning counterparts `Box` and `Ellipsoid`. Note that `HB` with transfer outperforms all methods shown in (a).

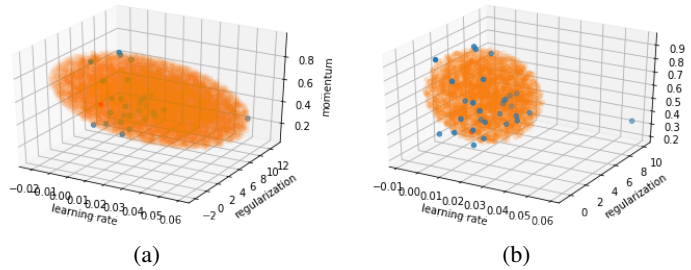

(a)                 (b)

Figure 2: Visualization of the learned `Ellipsoid` search space (a) without and (b) with slack variables. The blue dots are the observed evaluations and the orange dots are the samples drawn from the learned `Ellipsoid`. The slack-extension successfully excludes the outlier learning rate.

## 6.1 Tuning SGD for ridge regression

We consider the problem of tuning the parameters of stochastic gradient descent (SGD) when optimizing a set of 30 synthetic ridge regression problems with 81 input dimensions and 81 observations. The setting is inspired by [44] and is described in the Supplement. The HPO problem consists in tuning 3 HPs: learning rate in the range of $(0.001, 1.0)$, momentum in the range of $(0.3, 0.999)$ and regularization parameter in the range of $(0.001, 10.0)$. Figure 1a shows that the convergence to a good local minimum of conventional BO algorithms, such as `Random`, `GP`, and `SMAC`, is significantly boosted when the search space is learned (`Box`) from related problems. It is also interesting to note that all perform similarly once the search space is learned. The results for `GP warping` combined with `Box` are also similar, where the `Box` improves over `GP warping` but not as much as in the `GP` case due to significant performance gain with warping. We show the results in Supplement A.

The transfer learning methodology can be combined with resource-based BO algorithms, such as Hyperband [29]. We defined a unit of resource as three SGD updates (following [44]). By design, the more resources, the better the performance. Figure 1b shows that both the `Box` and `Ellipsoid`-based transfer benefit `HB`. Furthermore, `HB` with transfer is competitive with all other conventional BO algorithms, including model-based ones when comparing the RMSE across Figure 1a and Figure 1b.

We then studied the impact of introducing the slack variables to exclude outliers. One example of a learned `Ellipsoid` search space found for the 3 HPs of SGD on one of the ridge regression tasks is illustrated in Figure 2, together with its slack counterpart. The slack extension provides a more compact search space by discarding the outlier with learning rate value $\approx 0.06$. A similar result was obtained with the `Box`-based learned search space (see Supplement A).

## 6.2 Tuning binary classifiers over multiple OpenML data sets

We consider HPO for three popular binary classification algorithms: random forest (RF; 5 HPs), support vector machine (SVM; 4 HPs), and extreme gradient boosting (XGBoost; 10 HPs). Here, each problem consists of tuning one of these algorithms on a different data set. We leveraged OpenML [46],

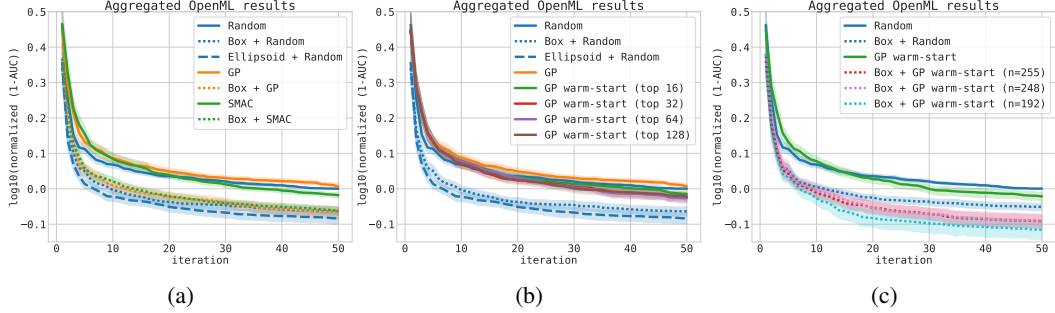

Figure 3: OpenML. (a) Performance of BO algorithms and their transfer learning counterparts. (b) Compares `GP warm-start` with `Box + Random` and `Ellipsoid + Random`. (c) Shows that `Box + GP warm-start` outperforms plain `GP warm-start`.

which provides evaluation data for many ML algorithm and data set pairs. Following [32], we selected the 30 most-evaluated data sets for each algorithm. The default search ranges were defined by the minimum and maximum hyperparameter value used by each algorithm when trained on any of the data sets. Results are shown in Figure 3a. The `Box` variants consistently outperform their non-transfer counterparts in terms of convergence speed, and `Box Random` performs en par with `Box SMAC` and `Box GP`. In this experiment, we also compared to `GP` with input warping [40], which exhibited a comparable boost when used in conjunction with the `Box` search space, slightly outperforming all other methods. Finally, `Ellipsoid Random` slightly outperforms `Box Random`. Next, we compare `Ellipsoid Random` and `Box Random` with two different transfer learning extensions of GP-based HPO, derived from [10]. Each black-box optimization problem is described by meta-features computed on the data set.[2] In `GP warm start`, the closest problem (among the 29 left out) in terms of $\ell_1$ distance of meta-features is selected, and its $k$ best evaluations are used to warm start the GP surrogate. Results are given in Figure 3b. Our simple search space transfer learning techniques outperform these GP-based extensions for all $k$. We also considered `GP warm start T=29`, which transfers the $k$ best evaluations from all the 29 left out problems, appending the meta-feature vector to the hyperparameter configuration as input to the GP (see Figure B1a in Supplement B). Results were qualitatively very similar, but the cubic scaling of GP surrogates renders `GP warm start T=29` unattractive for a large $T$ and/or $k$. In contrast, our transfer learning techniques are model-free.

In the next experiment, we combine our `Box` search space with these GP-based transfer techniques. In all methods, we use 256 random samples from each problem $t$: $n_t$ go to `Box`, the remaining $256 - n_t$ to `GP warm start`. The results are given in Figure 3c. It is clear that the `Box` improves plain `GP warm start` regardless of $n_t$. We also ran experiments with `GP warm start T=29 (n=*)` and `ABLR (n=*)` [32], a transfer HPO method which scales linearly in the total number of evaluations (as opposed to GP, which scales cubically). In all cases, `Box Random` is significantly outperformed by some `Box GP` or `Box ABLR` variant, demonstrating that additional gains are achievable by using some of the data from the related problems to tighten the search space (see Figure B1b and Figure B1c in Supplement B). Next, we studied the effect of the number $n_t$ of samples from each problem on `Ellipsoid Random`. Results are reported in Figure 4a. We found that with a small number ($n_t = 8$) of samples per problem, learning the search space already provided sizable gains. We also studied the effects of the number of related problems.[3] Results are given in Figure 4b. We see that our transfer learning by reducing the search space performs well even if only 3 previous problems are available, while transfering from 9 problems yields most of the improvements. The results for `Box Random` are similar (see Figure B2a- B2b in Supplement B).

Finally, we benchmark our slack variable extensions of `Box` and `Ellipsoid` from Section 5. As the number of related problems grows, the volume of the smallest box or ellipsoid enclosing all minima may be overly large due to some outlier solutions. For example, we observed that the optimal learning rate $\eta$ of XGBoost is typically $\leq 0.3$, except $\eta \approx 1$ for one data set. Our slack extensions are able to neutralize such outliers, reducing the learned search space and improving the performance (see Figure B3a and Figure B3b in Supplement B for more details).

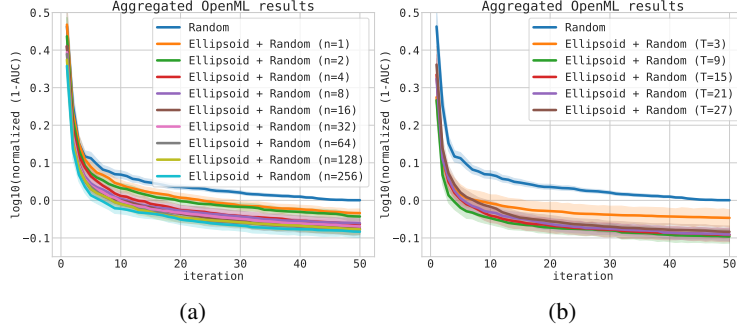

(a)                                    (b)

Figure 4: OpenML. (a) Sample size complexity and (b) robustness to the number of related problems for `Ellipsoid Random`.

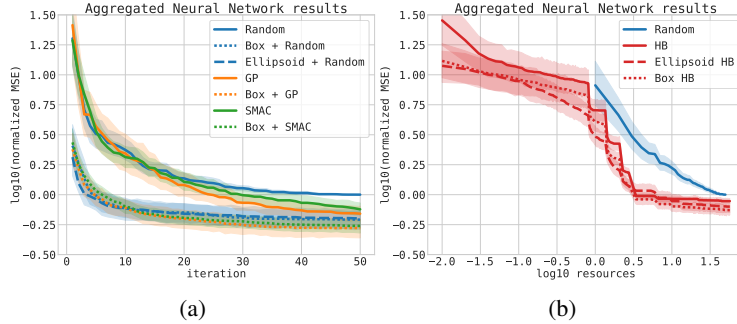

(a)                                    (b)

Figure 5: Feedforward neural network. (a) Performance of BO algorithms and their transfer learning counterparts. (b) Comparison with resource-based BO.

### 6.3 Tuning neural networks across multiple data sets

The last set of experiments we conduct consist of tuning the HPs of a 2-layer feed-forward neural network on 4 data sets [25], namely {`slice localization`, `protein structure`, `naval propulsion`, `parkinsons telemonitoring`}. The search space for this neural network contains the initial learning rate, batch size, learning rate schedule, as well as layer-specific widths, dropout rates and activation functions, thus in total 9 HPs. All the HPs are discretized and there are in total 62208 HP configurations which have been trained with ADAM with 100 epochs on the 4 data sets, optimizing the mean squared error. For each HP configuration, the learning curves (both on training and validation set) and the final test metric are saved and provided publicly by the authors [25]. As a result, we avoided re-evaluating the HPs, which significantly reduced our experiment time.

Each black-box optimization problem consists of tuning the neural network parameters over 1 data set after using 256 evaluations randomly chosen from the remaining 3 data sets to learn the search space. The default search space ranges are provided in [25]. We compared plain `Random`, `SMAC`, and `GP` to their variants based on `Box`. The results are illustrated in Figure 5a, where significant improvements can be observed. `Ellipsoid Random` also outperforms classic BO baselines such as `GP` and `SMAC`. Finally, Figure 5b demonstrates that `HB` can be further sped up by our transfer learning extensions.

These results also indicate that good solutions are typically found in the interior of the search space. To see how much accuracy could potentially be lost compared to methods that search over the entire search space, we re-ran the OpenML and neural network experiments using 16 times as many iterations (see Figure C1a and Figure C1b in Supplement C). Empirical evidence shows that excluding the best solution is not a concern in practice, and that restricting the search space leads to considerably faster convergence.

## 7 Conclusions

We presented a novel, modular approach to induce transfer learning in BO. Rather than designing a specialized multi-task model, our simple method automatically crafts promising search spaces

based on previously run experiments. Over an extensive set of benchmarks, we showed that our approach significantly speeds up the optimization, and can be seamlessly combined with a wide range of existing BO techniques. Beyond those we used in our experiments, we can further mention recent resource-aware optimizers [2, 8], evolutionary-based techniques [18, 34] and virtually any core improvement of BO, be it related to the acquisition function [48] or efficient parallelism [23, 47].

The proposed method could be extended in a model-based fashion, allowing us to *simultaneously* search for the best HP configurations $\mathbf{x}_t$'s for each data set, together with compact spaces containing all of these configurations. When evaluation data from a large number of tasks is available, heterogeneity in their minima may be better captured by employing a mixture of boxes or ellipsoids.

## Footnotes

[2] Four features: data set size; number of features; class imbalance; Naive Bayes landmark feature.

[3] For each of the 30 target problems, we pick $k < 29$ of the remaining ones at random, independently in each random repetition. We transfer $n_t = 256$ samples per problem.

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
