[Supplementary Material]

# Supplementary material
# Learning search spaces for Bayesian optimization: Another view of hyperparameter transfer learning

**Valerio Perrone, Huibin Shen, Matthias Seeger, Cédric Archambeau, Rodolphe Jenatton**[*]
Amazon
Berlin, Germany
{vperrone, huibishe, matthis, cedrica}@amazon.com

## A  Tuning SGD for ridge regression

The problem we consider in the toy SGD example is the following: denoting the squared loss by $\ell_i(\mathbf{u}) = \frac{1}{2}(\boldsymbol{\theta}_i^\top \mathbf{u} - \tau_i)^2$, we focus on solving

$$\min_{\mathbf{u} \in \mathbb{R}^p} L(\mathbf{u}) = \sum_{i=1}^{n} \{\ell_i(\mathbf{u}) + r\|\mathbf{u}\|^2\} \tag{1}$$

with the stochastic gradient (SGD) update rule at step $k$:

$$\mathbf{v} = \gamma \mathbf{v} + \eta \nabla L_i(\mathbf{u}^{(k)}) \tag{2}$$

$$\mathbf{u}^{(k+1)} = \mathbf{u}^{(k)} - \mathbf{v} \tag{3}$$

where $\boldsymbol{\theta}_i \in \mathbb{R}^p$ and $\tau_i \in \mathbb{R}$ refer to input and target of the regression problem while we use the momentum $\gamma \in [0.3, 0.999]$ together with the learning rate parametrization $\eta \in [0.001, 1.0]$ and $r \in [0.001, 10]$. The data $(\boldsymbol{\theta}_i, \tau_i)$'s are otherwise generated like in [2].

The complete results for this setting are presented in Figure A1. The `Box` and `Ellipsoid` approaches can significantly boost the baseline methods: `Random`, `GP`, `GP warping` and `SMAC`.

Figure A1: Tuning SGD for ridge regression. Comparison of BO algorithms with `Box` transfer learning counterparts.

We next studied the impact of introducing slack variables in the `Box` approach to exclude outliers. An example of a learned `Box` search space for the 3 hyperparameters of SGD on one of the ridge

---

[*]Work done while affiliated with Amazon; now at Google Brain, Berlin, rjenatton@google.com

regressions is illustrated in Figure A2, together with its slack counterpart. The slack extension effectively provides a more compact search space by leaving out the outlier learning rate value ($\sim$ 0.06). A similar result was obtained with the `Ellipsoid`-based learned search space.

Figure A2: Visualization of the learned `Box` search space (a) without and (b) with slack variables. The blue dots are the observed evaluations and the orange dots are the samples drawn from the learned `Box`. The slack-extension successfully exclude the outlier values for the learning rate.

# B    Tuning binary classifiers over multiple OpenML data sets

We first give more details on the experimental setup and then discuss some additional results not included in the main text. We follow the protocol and data collection from [1], which we describe below to provide a self-contained description.

## B.1    Experiment setup

In the OpenML [3] experiments, we considered the optimization of the hyperparameters of the following three algorithms:

- Support vector machine (SVM, `flow_id` 5891),
- Extreme gradient boosting (XGBoost, `flow_id` 6767).
- Random forest (RF. `flow_id` 6794).

Note that some hyperparameters can be log scaled. In Section D, we show an additional set of results for the log scaled search spaces.

### B.1.1    Support vector machine

The SVM tuning task consists of the following 4 hyperparameters:

- cost (float, min: 0.000986, max: 998.492437; can be log-scaled),
- degree (int, min: 2.0, max: 5.0),
- gamma (float, min: 0.000988, max: 913.373845; can be log-scaled),
- kernel (string, [linear, polynomial, radial, sigmoid]).

This tuning task exhibits conditional relationships with respect to the choice of the kernel.[2]

For this `flow_id`, we considered the 30 most evaluated data sets whose `task_ids` are: 10101, 145878, 146064, 14951, 34536, 34537, 3485, 3492, 3493, 3494, 37, 3889, 3891, 3899, 3902, 3903, 3913, 3918, 3950, 6566, 9889, 9914, 9946, 9952, 9967, 9971, 9976, 9978, 9980, 9983.

### B.1.2 XGBoost

The XGBoost tuning task consists of 10 hyperparameters:

- alpha (float, min: 0.000985, max: 1009.209690; can be log-scaled),
- booster (string, ['gbtree', 'gblinear']),
- colsample_bylevel (float, min: 0.046776, max: 0.998424),
- colsample_bytree (float, min: 0.062528, max: 0.999640),
- eta (float, min: 0.000979, max: 0.995686),
- lambda (float, min: 0.000978, max: 999.020893; can be log-scaled)
- max_depth (int, min: 1, max: 15; can be log-scaled),
- min_child_weight (float, min: 1.012169, max: 127.041806; can be log-scaled),
- nrounds (int, min: 3, max: 5000; can be log-scaled),
- subsample (float, min: 0.100215, max: 0.999830).

This tuning task exhibits conditional relationships with respect to the choice of the booster.[3]

For this `flow_id`, we considered the 30 most evaluated data sets whose `task_ids` are: 10093, 10101, 125923, 145847, 145857, 145862, 145872, 145878, 145953, 145972, 145976, 145979, 146064, 14951, 31, 3485, 3492, 3493, 37, 3896, 3903, 3913, 3917, 3918, 3, 49, 9914, 9946, 9952, 9967.

### B.1.3 Random forest

The random forest tuning task consists of the following 5 hyperparameters:

- mtry (int, min: 1, max: 36),
- num_tree (int, min: 1, max: 2000; can be log-scaled),
- replace (string, [true, false]),
- respect_unordered_factors (string, [true, false]),
- sample_fraction (float, 0.1, 0.99999)

For this `flow_id`, we considered the 30 most evaluated data sets whose `task_ids` are: 125923, 145804, 145836, 145839, 145855, 145862, 145878, 145972, 145976, 146065, 31, 3492, 3493, 37, 3896, 3902, 3913, 3917, 3918, 3950, 3, 49, 9914, 9952, 9957, 9967, 9970, 9971, 9978, 9983.

### B.2 Results

We first considered `GP warm-start T=29`, which transfers the $k$ best evaluations from each of the 29 left-out problems, appending the meta-feature vector to the hyperparameter configuration as input to the GP. The results are shown in Figure B1a. It is clear that `Box` and `Ellipsoid` combined with random already outperform this transfer learning baseline.

We also ran experiments with `Box + GP warm-start T=29 (n=*)` and `Box + ABLR (n=*)` [28], where $n$ evaluations from each related problem are used to find the bounding box, and $256 - n$ are used for GP and ABLR. In all cases, both `Box + Random` and the vanilla transfer learning approaches are significantly outperformed by `Box + GP warm-start` (in Figure B1b) and `Box + ABLR` (in Figure B1c). This demonstrates that alternative transfer learning algorithms can benefit from additional speed-ups when a subset of the available evaluations from the related problems are used to tighten the search space.

Next, we studied the effect of the number $n$ of samples from each problem on `Box + Random`. Results are reported in Figure B2a. We found that with a small number ($n = 8$) of samples per problem, learning the search space already provides sizable gains. We also studied the effects of the number of

Figure B1: OpenML. (a) Performance of `GP warm-start T=29` vs. our approaches. (b) `Box + GP warm-start T=29` vs. `GP warm-start T=29`. (c) `Box + ABLR` vs. `ABLR`.

source problems. Results are given in Figure B2b. By reducing the search space, our transfer learning approach performs well even if only 3 previous problems are available, while transferring from 9 problems yields most of the improvements. The results for `Ellipsoid + Random` are qualitatively similar.

Figure B2: OpenML. (a) Sample size complexity and (b) robustness the number of related problems for `Box + Random`.

We then studied the ability of the proposed slack-extensions of our method to remove outliers and lead to a more compact search space. As the number of related problems grows, the volume of the smallest box or ellipsoid enclosing all minimum points may be overly large due to some outlier solutions. For example, we observed that the optimal learning rate $\eta$ of XGBoost is typically $\leq 0.3$, except that $\eta \approx 1$ for one data set. Our slack extensions are able to neutralize such outliers, as shown in Figure B3a for an example 2-d slice of the slack-ellipsoid. On the XGBoost problems, the slack formulations are able to appropriately shrink the search space and considerably improve performance (Figure B3b). The performance gain does not emerge for SVM in Figure B4a and RF in Figure B4b, which can be attributed to the available solutions being more homogeneous across problems.

## C  Exploring the search space with more resources

The approach we propose aims to lift the burden of choosing the search space. When running experiments for a fixed amount of iterations, this has the effect of exploring the (restricted) search space more densely, which we showed to be beneficial. To assess how much accuracy can potentially be lost compared to methods that search the entire space (with more resources), we re-ran our experiments with 16 times as many iterations. Figure C1a and Figure C1b respectively show the OpenML and neural network results aggregated across all tasks: restricting the search space leads to considerably faster convergence and `GP` fails to find a better solution, only eventually catching up in the neural network case. This suggests that there is almost no loss of performance, but just a speed-up effect of finding a very good solution in as few evaluations as possible.

Figure B3: XGBoost. (a) The fitted slack-ellipsoid effectively removes the outlier step size eta. (b) The slack extensions further boost performance over the vanilla box and ellipsoid formulations.

Figure B4: Results for the slack extensions on random forest (RF) (a) and SVM (b).

## D  Tuning OpenML binary classifiers in the log scaled search space

We presented results on the OpenML data in the main text and Section B of the supplementary material. One feature of the OpenML setting is that the hyperparameter search spaces can be wide, such as the n_rounds of XGBoost which can go up to $5000$. Therefore, we re-ran the OpenML experiments in the alternative scenario of log-scaled search spaces. A key message is that the conclusions are qualitatively similar, while the gaps between the different models are smaller since the tasks are overall simpler.

We repeated the experiments on `Box` combined with `Random, GP, SMAC` and show the results in Figure D1a, where we see consistent improvements using the `Box` approach. Comparing to the vanilla transfer learning method, namely `GP warm-start`, `Box + GP` also demonstrated improved performance as shown in Figure D1b. Crucially, we can combine `GP warm-start` with `Box` to achieve even better performance as shown in Figure D1c.

We then studied the effects of the number of samples and number of related problems used to learn the search space of `Ellipsoid + Random` and `Box + Random`. Both the `Ellipsoid` and `Box` are quite robust to the number of samples: the difference is smaller than in the setting without log scaling, and they both improve over `Random`, as shown in Figure D2a and Figure D3a. Figure D2b and Figure D3b show that, while using 3 related problems seems not enough to learn a good search space for the two approaches, with 9 related problems both the `Box` and the `Ellipsoid` improve on `Random`, especially at the beginning of the optimization. Finally, the `Ellipsoid` approach tends to outperforms `Box`, pointing to the benefits of a more flexible representation of the search space.

Figure C1: Results with 16 times more iterations on OpenML algorithms (a) and a 2-layer neural network (b).

Figure D1: OpenML in log scaled search space. (a) Performance of BO algorithms and their transfer learning counterparts. (b) `Box`, `Ellipsoid` vs. `GP warm-start`. (c) `Box + GP warm start` vs. `GP warm start`.

Figure D2: OpenML in log scaled search space: (a) Sample size complexity and (b) robustness to the number of related problems for `Ellipsoid Random`.

Figure D3: OpenML in log scaled search space: (a) Sample size complexity and (b) robustness to the number of related problems for `Box Random`.

## Footnotes

[2]For details, we refer the reader to the API from `www.rdocumentation.org/packages/e1071/versions/1.6-8/topics/svm`

[3]For details, we refer the interested readers to the API from www.rdocumentation.org/packages/xgboost/versions/0.6-4