[Reviews · NeurIPS 2019]

Reviewer 1



* The authors seem unaware of prior work on learning which hyperparameters are most important to tune and which values are most promising. For instance: https://arxiv.org/pdf/1710.04725.pdf learns priors over the hyperparameter space. * The method only works for completely numeric spaces. The numeric and non-numeric subspaces could be tuned separately, but then you can't learn interactions between both kinds of features, while a full Bayesian optimization approach could. * The results show excellent anytime performance: with limited resources, it provides good solutions fast. However, since the search space is restricted, it is possible that the very best configurations fall outside the search space, and that a more extensive search would yield a better model. This is not demonstrated since all experiments were cut off after 50 iterations. * the ellipsoid introduces an extra hyper-hyperparameter. How should it be tuned? * Generalizability is questionable. The toy problem is really small, I don't really see what to take away from these results. The OpenML experiments are larger (30 datasets) and up to 10 hyperparameters, but this is still rather small to other work in the literature, e.g. autosklearn with hundreds of hyperparameters and datasets. The experiments all included very densely sampled problems, it is unclear how accurate this method is if only sparse prior metadata is available. One sign that is troubling is that randomsearch seems to outperform more sample-efficient approaches easily in these experiments. * Why were only 4 meta-features used? Most prior work uses many more. * In is unclear why the hyperparameters were discretized in the neural network experiments. Update after the author feedback: Most of my concerns have been addressed. I still have some reservations about generalizability but the current experiments are promising.

Reviewer 2



[I have read the author rebuttal and upgraded my score. My concern about generalization still remains, and I hope the authors can devote maybe a sentence or two to it in the final draft - even something to the effect of "it is a concern; experimental evidence suggests it is not a great concern."] Summary: For any given ML algorithm, e.g., random forests, the paper proposes a transfer-learning approach for selection of hyperparameters (limited to those parameters that can be ordered) wherein a bounding space is constructed from previous evaluations of that algorithm on other datasets. Two types of bounding spaces are described. The box space is the tightest bounding box covering the best known hyperparameter settings for previous datasets. The ellipsoid is found as the smallest-volume ellipsoid covering the best known settings (via convex optimization). The box space can be utilized by both model-free Bayesian optimization (BO), e.g., random search or Hyperband, and model-based BO, e.g., GP, whereas the ellipsoid space can only be utilized by model-free BO. I liked the idea of transforming the problem of how to leverage historical performance for hyperparameter selection into one of how to learn an appropriate search space. The paper demonstrates that a relatively simple dataset-independent procedure provides performance gains over not transferring. This is also what I'm concerned about with the approach: what if a dataset itself is an outlier to previous datasets? In this case, why would we expect "successful" hyperparameter ranges on previous datasets to bound the right setting for the new dataset? Using the box space could be disastrous in this setting (while it could be argued that transfering does not help in this case, it could also be argued that this would not be known a priori). This leads me to believe the approach is not robust in its current form. The ellipsoid space may be more resilient in this regard, but it is limited to model-free methods. It would be better if the search space learning could incorporate dataset information such that graceful performance degradation would occur when attempting to transfer to a unique dataset for which historical performance fails to inform. I do think the finding that certain hyperparameter ranges for common ML algorithms should be tried first in hyperparameter selection is a good empirical finding, but not significant enough on its own. Originality: This reviewer is not aware of a similar proposal, but is not familiar with all the literature that falls into this category. Quality: The experimental section (and supplement) considered a large range of settings for the problem. Clarity: The authors have done a great job wrt/ organization, exposition, and results descriptions. There was only one place where I felt confused in an otherwise very readable paper. Does the experimental parameter n in lines 256-257 correspond to n_t in line 83? If not, please provide more detail on what it represents in the experiment. If so, how were the previous evaluations selected, and what is T for this experiment?

Reviewer 3



Summary of the main ideas: A really novel methodology to learn search spaces in Bayesian Optimization that is able to build ellipsoidal search spaces that are sampled through rejection sampling. I like this paper due to the fact that it seems to me after reading several papers about this topic this this methodology is the best, and most practical, to learn the search space in Bayesian Optimization. It has further work in the sense that can not be applied as it is to GPs but I think that using an approach such as for example PESMOC to model the ellipsoid as a constrained solves it easily (it is commented in the paper). I see it so clearly that I would even be willing to collaborate on that idea. The approach gains quality in the sense that it does not need parameters (critical in BO) in proposes several methods and they are simple yet effective. Brilliant paper from my point of view. Related to: -> Learning search spaces in BO. Strengths: Effective and simple methods that solve a popular task in BO more easily than previous methods. Proposes several alternatives. Proposes further work to address the only tasks that are pending. It seems to me that opens a new set of methodogies to be further improved. The paper is very well written. Weaknesses: We can argue that is not a complete piece of work, but it is veery rigurous. Lack of theoretical content, but it does not matter as the empirical content and the proposed methodologies are solid. Does this submission add value to the NIPS community? : Yes it does. I miss simple but effective methods for BO and this paper contains them. It contains further work to be done. It is an approach that I would use if I were working in a company, in the sense that it is easy and pragmatic. Quality: Is this submission technically sound?: Yes it is. Estimating the search space as an optimization problem and with different geometrical shapes in the sense that the paper performs is sound. Are claims well supported by theoretical analysis or experimental results?: Experimental results support the claims. Is this a complete piece of work or work in progress?: It is a work in progress though, GPs are not covered and are the most useful model in BO. But the path is set in this work. Are the authors careful and honest about evaluating both the strengths and weaknesses of their work?: I think so from my humble opinion. Clarity: Is the submission clearly written?: Yes it is, it is even didactic. Good paper that I would show to my students. Is it well organized?: Yes it is. Does it adequately inform the reader?: Yes it does. Originality: Are the tasks or methods new?: They are tools used in other problems but their application in BO is sound. Is the work a novel combination of well-known techniques?: Combination of well-known techniques, very well done. Is it clear how this work differs from previous contributions?: Yes it does. Is related work adequately cited?: Yes I think so. Significance: Are the results important?: I think so, they seem to speed up the optimization. Are others likely to use the ideas or build on them?: Yes, definitely, I would do it and I am thinking in contacting them after publication or rejection is done, I found this piece of work very interesting. Does the submission address a difficult task in a better way than previous work?: Yes absolutely. Does it advance the state of the art in a demonstrable way?: Yes it does. Does it provide unique data, unique conclusions about existing data, or a unique theoretical or experimental approach?: Yes, experiments provide good results. Arguments for acceptance: Effective and simple methods that solve a popular task in BO more easily than previous methods. Proposes several alternatives. Proposes further work to address the only tasks that are pending. It seems to me that opens a new set of methodogies to be further improved. The paper is very well written. Arguments against acceptance: Honestly, I would accept this paper, I know that NIPS has high standards, but I think that it is a good paper.

[Author Response · NeurIPS 2019]

We thank the reviewers for the positive and constructive comments. Our detailed responses are below.

**R2**. We are grateful to the reviewer for the comments and suggested reference, which we will discuss in the revision. The idea of learning hyperparameter importance bears interesting connections to our work, and is an alternative viewpoint on transfer learning that could be combined with our novel representations of the search space. It would be insightful to apply a post-hoc functional ANOVA analysis to validate the way we prune the search space a priori.

We would like to clarify that, while the first step of our method only restricts the search space for numerical features, it still performs standard BO over the joint space, capturing the interactions between the two types of variables. The bounding box formulation naturally handles categorical parameters that are one-hot encoded: if none of the $\mathbf{x}_t^\star$'s uses a given category, say corresponding to the $j$-th dimension, we have $[\mathbf{x}_t^\star]_j = 0$ for all $t$, and the corresponding entries of $\mathbf{l}^*$ and $\mathbf{u}^*$ in (4) will satisfy $l_j^* = u_j^* = 0$; this means that a category not present in the $\mathbf{x}_t^\star$'s is automatically pruned out of the search space. We did not mention this aspect to keep the discussion simpler, but would be happy to clarify this in the revision. We would also like to clarify that there are no extra hyperparameters introduced with the ellipsoid (see equation (6) of the paper), similar to the bounding box case. Providing a parameter-free, off-the-shelf methodology was one of the main motivations of our approach.

Aggregated Neural Network results

Our goal was to lift the burden of choosing the search space. When running experiments for a fixed amount of resources as we do, this has the effect of exploring the (restricted) search space more densely, which we showed to be beneficial. Following on the reviewer's observations, we re-ran our experiments with 16 times as many iterations to assess how much accuracy can potentially be lost compared to methods that search the entire space (with more resources). The figures on the right show the neural network and OpenML results: restricting the search space leads to considerably faster convergence and the GP fails to find a better solution, only eventually catching up in the neural network case. This suggests that there is almost no loss of performance, but just a speed up effect of finding a very good solution in as few evaluations as possible. We thank the reviewer for this insightful comment and would like to add these new results to the revision.

Aggregated OpenML results

*Toy SGD experiment*: We replicated the setup of Valkov et al., 2018, to provide insights into the proposed algorithm in a fully controlled setting, where resource-aware optimizers can be tested and we could build intuition by visualizing the evaluations together with the search space restriction in Fig. 2 of the paper and confirming a meaningful behavior. *Neural network experiment*: Following the setup of Klein and Hutter (2019), the parameters in the neural network experiment were discretized to allow for an exhaustive look-up table, eliminate the noise coming from interpolation and ease reproducibility. *Larger hyperparameter setting*: We agree that finding automatic search spaces in larger hyperparameter settings would be a valuable study to conduct, with the potential of further speeding up the BO by removing or restricting irrelevant dimensions. As we aimed to compare to previous transfer learning approaches developed in similar settings, we chose to benchmark the moderate-sized P (around 10) regime as in Snoek, et al. (2015), Springenberg, et al. (2016), and Perrone et al. (2018).

**R3**. Choosing the search space is difficult in practice and being conservative, by picking a large search space, can negatively impact the optimization performance. Our goal was to highlight this aspect as it is often overlooked in the literature, and to propose a simple, yet effective methodology to automate this critical step. When evaluating the methodology we asked ourselves the same question regarding the possibility of excluding the best solution in some situations. The experimental results show that focusing on the search space to induce transfer learning in BO is more effective than more complex and computationally intensive approaches. We also found that, when we were restricting the search space, good solutions were always found in the interior of the search space. We conducted an extra set of experiments to see how much accuracy could potentially be lost compared to methods that search the entire space (with more resources). The results indicate that results were robust and the difference was small, if any (see Figs on the side).

We confirm $n$ in lines 256-257 corresponds to $n_t$ in line 83, which we will fix in the revision. In all OpenML experiments, the $n_t$ previous evaluations per task are drawn uniformly at random from the ones available, with $T = 30$ in all OpenML experiments except for Figure 4b.

**R4**. We would like to thank the reviewer for the positive feedback and suggestions. Several extensions are indeed possible, some of which are outlined in the discussion section. We thank the reviewer for the additional suggestion of combining our method with PESMOC and would be happy to further discuss this off-line. Striving for reproducibility, we made the algorithmic details as thorough as possible, and provided the pseudo-code of the proposed methodology.

[Meta-Review · NeurIPS 2019]

The reviewers appreciated the simplicity of the approach and the rigorous evaluation. They recommended discussing the generalizability of the approach in one or two sentences in the final version of the paper.